


# The Kinetic Energy Budget of the Atmosphere (KEBA) model 1.0: A simple yet physical approach for estimating regional wind energy resource potentials that includes the kinetic energy removal effect by wind turbines

Axel Kleidon[1] and Lee M. Miller[2]

[1]Biospheric Theory and Modelling Group, Max-Planck-Institut für Biogeochemie, Jena, Germany
[2]Atmospheric and Environmental Research, Lexington, MA, USA

*Correspondence to:* Axel Kleidon (akleidon@bgc-jena.mpg.de)

**Abstract.** With the current expansion of wind power as a renewable energy source, wind turbines increasingly extract kinetic energy from the atmosphere, thus impacting its energy resource. Here we present a simple, physics-based model (KEBA) to estimate wind energy resource potentials that explicitly account for this removal effect. The model is based on the regional kinetic energy budget of the atmospheric boundary layer that encloses the wind farms of a region. This budget is shaped by hor-
izontal and vertical influx of kinetic energy from upwind regions and the free atmosphere above as well as the energy removal by the turbines, dissipative losses due to surface friction and wakes, and downwind outflux. These terms can be formulated in a simple, yet physical way, yielding analytic expressions for how wind speeds and energy yields are reduced with increasing deployment of wind turbines within a region. We show that KEBA estimates compare very well to the modelling results of a previously published study in which wind farms of different sizes and in different regions were simulated interactively with the
WRF atmospheric model. Compared to a reference case without the effect of reduced wind speeds, yields can drop by more than 50% at scales greater than 100 km, depending on turbine spacing and the wind conditions of the region. KEBA is able to reproduce these reductions in energy yield compared to the simulated climatological means in WRF ($n = 36$ simulations; $r^2 = 0.822$). The kinetic energy flux diagnostics of KEBA show that this reduction occurs because the total yield of the simulated wind farms approaches a similar magnitude as the influx of kinetic energy. Additionally, KEBA estimates the slowing of
the region's wind speeds, the associated reduction in electricity yields, and how both are due to the depletion of the horizontal influx of kinetic energy by the wind farms. This limits typical large-scale wind energy potentials to less than 1 W m$^{-2}$ of surface area for wind farms with downwind lengths of more than 100 km, although this limit may be higher in windy regions. This reduction with downwind length makes these yields consistent with GCM-based idealized simulations of large-scale wind energy resource potentials. We conclude that KEBA is a transparent and informative modelling approach to advance the sci-
entific understanding of wind energy limits, and can be used to estimate regional wind energy resource potentials that account for the depletion of wind speeds.





## 1   Introduction

The use of wind energy as a renewable energy resource has substantially increased over the last decades in the attempt to decarbonize the energy system. Particularly wind over the sea is seen as a tremendous, yet underutilized energy resource. In Europe alone, the current installed capacity of 22 GW in offshore wind power has increased by 3.5 GW in 2019 (WindEurope,

2019a). It is expected to expand further to 450 GW and more by 2050 (WindEurope, 2019b), playing a key role in Europe's transition to a carbon neutral energy system by 2050.

There is, however, a substantial discrepancy in how efficient wind turbines are in generating electricity, depending on the scale of deployment. An isolated turbine in an offshore environment with high, continuous wind speeds may generate electricity highly efficiently, with a capacity factor (i.e., the ratio of generated electricity to the capacity of the turbine) above 50% and

more than 4300 full load hours per year. These high efficiencies are typically used in assessments of offshore wind resource potentials (e.g. WindEurope, 2019b). However, the more wind turbines that are deployed within a region, the more these remove kinetic energy from the atmosphere, leaving less behind, resulting in lower wind speeds and lower efficiencies of turbines downwind. Idealized climate model simulations at the planetary scale showed that this wind depletion effect results in much lower large-scale limits to wind power (Miller et al., 2011; Jacobson and Archer, 2012; Adams and Keith, 2013; Miller

and Kleidon, 2016) in the order of about 1 W m$^{-2}$ of surface area or less. The resulting wind energy potentials are then below the rate by which the natural atmosphere dissipates kinetic energy near the surface. It is this effect on wind speeds that results in a decline in turbine efficiencies when deploying wind energy at increasingly larger scales. Regional simulations with weather forecasting models have shown similar effects in hypothetical simulations (Adams and Keith, 2013; Miller et al., 2015; Volker et al., 2017). What this demonstrates is that as wind energy use expands to larger scales, turbine efficiency becomes less a

question of the technology being used and more about how the natural, atmospheric environment supplies the kinetic energy extracted by the wind farms.

Here we describe a modelling approach to estimate regional-scale wind energy resource potentials that explicitly accounts for the wind speed reductions and lower yields. The goal of this modelling approach is to provide simple and transparent, first-order estimates based on physical concepts. To do so, we focus on the Kinetic Energy Budget of the boundary layer in the lower

Atmosphere (KEBA), as shown in Fig. 1. We consider the volume of the atmosphere that encloses the region in which wind farms are deployed and that extends to the height of the atmospheric boundary layer. The boundary layer is used here as the basis for our budgeting, as it represents the region of the lower atmosphere that is typically considered to be well mixed, so that when turbines remove kinetic energy, this mixing replenishes the kinetic energy in the flow behind the turbines. By balancing the fluxes of kinetic energy of this volume, which include the energy extracted by the wind turbines as one of the terms, we

obtain an analytic approach to formulate the decline in wind speeds with greater wind energy use. Note that this approach extends beyond turbine wakes, the immediate reduction of wind speeds right behind individual turbines in wind farms. Turbine wakes cause reductions in yields of downwind turbines due to the incomplete replenishment of the kinetic energy from the surrounding flow. This is an effect that has been well observed and modelled (e.g. Frandsen et al., 2006; Barthelmie et al., 2010; Emeis, 2010). We aim here at a broader description, not focussing on individual turbines and incomplete mixing, but



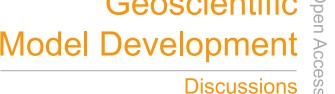

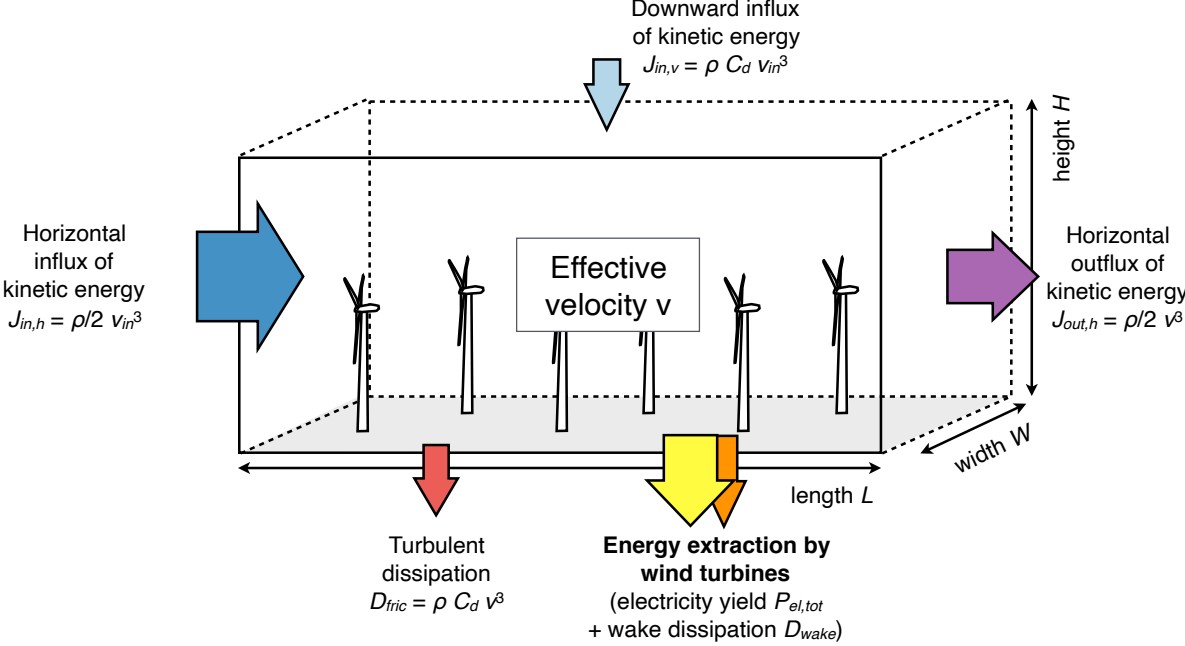

**Figure 1.** Illustration of the Kinetic Energy Budget of the Atmosphere (KEBA) approach to estimate regional wind energy potentials, which considers the fluxes of kinetic energy in and out of a virtual volume that encloses a given region of dimensions $L \cdot W \cdot H$ to infer limits to electricity generation as well as wind speed reductions. The mathematical formulations of the fluxes are indicated at the arrows.

rather at the cumulative effects of all wind turbines within a region in depleting the kinetic energy of the boundary layer. We also aim for a first-order estimate that may not be as precise, but is sufficiently simple so that it can be implemented in a way that it is accessible to a wider range of scientists and can provide a transparent way based on physical concepts to estimate

regional wind energy potentials.

   Naturally, our KEBA approach needs to be tested to see whether it can reasonably reproduce the effects and magnitudes simulated by far more complex simulation models. To do this, we use the published results of numerical simulations performed by Volker et al. (2017). Their study used a wind farm parameterization and the WRF atmospheric model to evaluate the effects of wind farms of different sizes, different turbine spacings, and in different regions with respect to their yields. As their study

represents a broad range of sensitivities and the publication includes the necessary information to evaluate against, we use this study to evaluate how well KEBA can estimate yields of wind energy across various scales.

   In the following, we first describe the mathematical formulation of KEBA in section 2, where we describe the kinetic energy budget and how wind speed reductions as well as turbine yields are simultaneously derived from KEBA from the wind forcing of the region. The resulting equations of KEBA can easily be implemented in a spreadsheet, which is included in this paper

as supplementary material. We then describe its evaluation using the simulations by Volker et al. (2017) in section 3. There, we also show the utility of diagnosing the kinetic energy budget to understand why turbine yields need to decline at larger





deployment scales. The sensitivity of the model to meteorological conditions, in terms of boundary layer height as well as the drag coefficient, is evaluated as well. We then use the sensitivity of KEBA to downwind length of a wind farm to illustrate the scale at which the average turbine yields approach the large-scale wind energy limit. We briefly discuss some potential

limitations of the approach and close with a brief summary and conclusions.

## 2 Model formulation

### 2.1 Overview

The goal of the Kinetic Energy Budget of the Atmosphere (short KEBA) model is to provide a simple and transparent, yet physically-based approach to estimate wind energy potentials for a given region across scales that can reproduce the wind

speed reductions found in much more complex numerical simulation models. It uses an observed record of wind speeds, dimensions of the region, turbine characteristics as well as the number of turbines as an input. It predicts the reduction in wind speeds as well as the generated yields as output. The simplicity of the approach allows for it to be implemented in a spreadsheet, which is provided as Supplemental Material.

KEBA derives an effective wind speed within a region of wind turbines from the different fluxes that add, remove or dissipate

kinetic energy within the associated atmospheric air volume that encloses the region (Fig. 1). For this, KEBA uses information of the unaffected wind speed, $v_{in}$, of the region in combination with a few meteorological parameters (the drag coefficient $C_d$ and a typical boundary layer height $H$) as well as turbine characteristics (number of turbines, $N$, rated capacity, $P_{el,max}$, rotor-swept area, $A_{rotor}$, power coefficient, $\eta$, as well as cut-in and cut-out velocities, $v_{min}$ and $v_{max}$). The enclosing atmospheric volume is described by the dimensions of the cross section perpendicular to the wind direction (height $H$ and width $W$), and

the downwind depth $L$ of the considered region. The variables are summarized in Table 1.

The effective wind speed $v$ within the region is derived from the kinetic energy budget of the enclosing atmospheric volume. The influx of kinetic energy, $J_{in,h}$, through the upwind cross section ($HW$) and the vertical downward mixing $J_{in,v}$ over the area ($WL$) of the region add kinetic energy (dark and light blue arrows in Fig. 1), while the electricity generation, or yield, of the wind turbines, $P_{el,tot}$ (yellow arrow), the outflux of kinetic energy, $J_{out,h}$, downwind of the region (purple arrow),

and dissipative losses by surface friction, $D_{fric}$ (red arrow), and wake turbulence, $D_{wake}$ (orange arrow), remove or dissipate kinetic energy. We neglect changes in kinetic energy within the region and dissipative losses by mixing taking place above the wind farms but inside the air volume. The balance of these fluxes is given by

$$J_{in,tot} = J_{in,h} + J_{in,v} = J_{out,h} + P_{el,tot} + D_{fric} + D_{wake} \tag{1}$$





**Table 1.** Overview of KEBA variables and how these are specified or computed.

| Symbol | Description | Units (or value) | Comment |
|---|---|---|---|
| $v_{in}$ | Wind speed (unaffected by wind turbines) | m s$^{-1}$ | forcing |
| $v$ | Wind speed within region (affected by wind turbines) | m s$^{-1}$ | Eq. 8 |
| $\rho$ | Air density | kg m$^{-3}$ | $\approx 1.1$ kg m$^{-3}$ |
| $H$ | Boundary layer height | m | prescribed |
| $C_d$ | Drag coefficent | - | prescribed |
| **Turbine characteristics** | | | |
| $P_{el,max}$ | Turbine capacity | W | specified |
| $D$ | Rotor diameter | m | specified |
| $A_{rotor}$ | Rotor-swept area | m$^2$ | $\pi(D/2)^2$ |
| $\eta$ | Power coefficient for $v_{min} \leq v \leq v_{rated}$ | - | specified |
| $v_{min}$ | Cut-in velocity | m s$^{-1}$ | specified |
| $v_{rated}$ | Rated velocity | m s$^{-1}$ | $P_{el,max} = \rho/2 \cdot v_{rated}^3 \cdot \eta A_{rotor}$ |
| $v_{max}$ | Cut-out velocity | m s$^{-1}$ | specified |
| **Scenario characteristics** | | | |
| $N$ | Number of turbines | - | specified |
| $W$ | Width of cross section of wind farm region | m | specified |
| $L$ | Downwind length of wind farm region | m | specified |
| $n$ | Turbine number density | m$^{-2}$ | $N/(WL)$ |
| **Energy fluxes** | | | |
| $J_{in,tot}$ | Total influx of kinetic energy | W | Eq. 1 |
| $J_{in,h}$ | Horizontal influx of kinetic energy | W | Eq. 2 |
| $J_{in,v}$ | Vertical mixing of kinetic energy | W | Eq. 3 |
| $P_{el,tot}$ | Electrical power generation, or yield, of all turbines | W | Eq. 10 or 12 |
| $D_{fric}$ | Frictional dissipation | W | Eq. 6 |
| $D_{wake}$ | Dissipation in turbine wakes | W | Eq. 7 |
| $J_{out,h}$ | Horizontal outflux of kinetic energy | W | Eq. 5 |





## 2.2 Energy fluxes

The total influx of kinetic energy, $J_{in,tot}$, is described in terms of the upwind wind speed $v_{in}$ by the horizontal influx of kinetic energy by the wind through the cross-sectional area $WH$,

$$J_{in,h} = WH \cdot \frac{\rho}{2} v_{in}^3 \tag{2}$$

and by the vertical mixing due to surface friction over the surface area $WL$,

$$J_{in,v} = WL \cdot \rho C_d v_{in}^3 \tag{3}$$

where $C_d$ is the drag coefficient of the surface. The use of the surface drag coefficient is used here as an approximation.

The loss terms of kinetic energy are described with respect to an effective wind speed, $v$, within the region. For simplicity, we derive an effective velocity $v$ that is representative for the mean generation of wind turbines, neglecting variations of wind speed within the region, particularly in the downwind direction.

For the electricity generation, or yield, $P_{el,tot}$, $N$ wind turbines of the same characteristics are being considered, with each turbine having a rated capacity of $P_{el,max}$. The turbines have a rotor-swept cross sectional area $A_{rotor}$ and a power coefficient
$\eta$. The yield, $P_{el,tot}$, is then described by

$$P_{el,tot} = N \cdot \min \left[ \eta A_{rotor} \cdot \frac{\rho}{2} v^3 ; P_{el,max} \right] \tag{4}$$

with $\rho$ being the air density and $v$ the effective wind speed. The minimum function is being used with the two arguments inside the parentheses to distingish the case when the turbines operate below or at their capacity. In the case that the wind speed is below the cut-in velocity ($v_{in} \leq v_{min}$) or above the cut-out velocity ($v_{in} \geq v_{max}$), no generation is assumed ($P_{el,tot} = 0$),
resulting in no effect on the wind speed.

The outflux of kinetic energy, $J_{out,h}$, downwind of the region is described by

$$J_{out,h} = WH \cdot \frac{\rho}{2} v^3 \tag{5}$$

Dissipation by surface friction, $D_{fric}$, is described by a typical surface drag parameterization of the form

$$D_{fric} = WL \cdot \rho C_d \cdot v^3 \tag{6}$$

where $C_d$ is the drag coefficient of the surface, which can be calculated for neutral conditions using the roughness length, $z_0$, of the surface by $C_d = \kappa^2 / \ln^2(z/z_0)$, where $\kappa \approx 0.4$ is the von-Karman constant and $z$ the reference height at which the wind speed is being measured.

Dissipation of kinetic energy by wake turbulence, $D_{wake}$, caused by the wind turbines is assumed to be half of the generated electricity:

$$D_{wake} = \frac{1}{2} \cdot P_{el,tot} \tag{7}$$

This simple approximation is based on theoretical work by Corten (2001). Dissipative losses by the downward mixing of kinetic energy are neglected.





## 2.3 Estimation of wind speed and yields

The equations 1 - 7 are combined to derive an expression for the effective wind speed, $v$,

$$v = f_{red}^{1/3} \cdot v_{in} \tag{8}$$


where $f_{red}$ is a reduction factor that depends on the characteristics of the enclosing air volume and the installed capacity of the region. In the case in which the turbines of the region operate below their rated capacity, this reduction factor is given by

$$f_{red} = \frac{H + 2C_d L}{H + 2C_d L + \frac{3}{2} n \eta A_{rotor} L} \tag{9}$$

where $n = N/(WL)$ is the turbine number density.

The yield of the wind farm, $P_{el,tot}$, is then given by

$$P_{el,tot} = f_{red} \cdot N \cdot \eta A_{rotor} \cdot \frac{\rho}{2} v_{in}^3 \tag{10}$$

which is the same as Eq. 4, except for the use of $f_{red}$ and $v_{in}$ instead of $v$. In other words, the reduction in yield in this formulation is captured entirely by the factor $f_{red}$. A value of $f_{red} = 1$ represents the case of an isolated wind turbine in which wind speeds are unaffected. The lower the value of $f_{red}$ is, the greater the reduction in effective wind speed and in yield. The

primary factor that results in a reduction is the number of turbines, $N$, combined with rotor cross-sectional area, $A_{rotor}$, as this reduces the value of $f_{red}$ in Eq. 9.

In the case in which the wind farm operates at its rated capacity ($v_{in} \geq v_{rated}$), the reduction factor takes a different form of

$$f_{red} = 1 - \frac{3}{2} \cdot \frac{1}{H + 2C_d L} \cdot \frac{H}{L} \cdot \frac{N \cdot P_{el,max}}{J_{in,h}} \tag{11}$$

where the last fraction on the right hand side is the ratio of total installed capacity in the region divided by the total horizontal influx of kinetic energy. The electricity generation simply takes the form of

$$P_{el,tot} = N \cdot P_{el,max} \tag{12}$$

## 2.4 Diagnostic energy fluxes

To link and visualise reductions in yield and wind speeds, it is instructive to explicitly look at the fluxes that shape the kinetic

energy balance. The influxes of kinetic energy, $J_{in,h}$ and $J_{in,v}$, are given by Eqs. 2 and 3, respectively. The yield of the wind turbines is given by the flux $P_{el,tot}$ (Eqs. 10 or 12, depending whether $v_{in} < v_{rated}$ or $v_{in} \geq v_{rated}$), with dissipation $D_{wake}$ by wake turbulence given by Eq. 7. The remaining two terms, dissipation by surface friction, $D_{fric}$, and the outflux of kinetic energy, $J_{out,h}$, can then be obtained from the energy balance, Eq. 1, and take the simplified forms of

$$D_{fric} = \frac{2C_d L}{H + 2C_d L} \cdot (J_{in,tot} - P_{el,tot} - D_{wake}) \tag{13}$$



and

$$J_{out_h} = \frac{H}{H + 2C_d L} \cdot (J_{in,tot} - P_{el,tot} - D_{wake}) \tag{14}$$

These two equations are a reformulation of Eqs. 5 and 6. They simplify the calculation of the budget because these can be directly derived from the forcing, $J_{in,tot}$, and the rate of electricity generation, $P_{el}$, and no distinction needs to be made whether turbines operate below or at capacity as this is already accounted for in $P_{el,tot}$.

### 2.5  Model implementation

Equations 8 - 12 describe the KEBA approach. These equations describe the lower wind speed within the region (Eq. 8) as a function of the reduction factor $f_{red}$ (either given by Eq. 9 or 11) and the meteorological forcing given by $v_{in}$. The generated yield within the region is then described by Eq. 10 or 12. Note that these expressions are very similar to well established

formulations, particularly regarding the yield. However, instead of a fixed reduction factor to account for wake effects in wind farms, the reduction factor in Eq. 10 is not a fixed value and it is also not empirically determined. Instead, the reduction factor depends explicitly on the size of the region (width $W$ and downwind length $L$ of the region) as well as on the number of wind turbines and their characteristics (rated capacity $P_{el,max}$, power coefficient $\eta$, number of turbines $N$, rotor-swept area $A_{rotor}$, cut-in and cut-out velocities), but also on meteorological characteristics (boundary layer height $H$, drag coefficient $C_d$). The

KEBA approach is thus mostly based on the physical concept of a kinetic energy balance and it requires comparatively little empirical parameters to infer the magnitude of wind speed and yield reduction with certain installed capacities at the regional scale.

To estimate wind energy yields, KEBA needs meteorological input in form of wind speeds, $v_{in}$, the height of the boundary layer, $H$, and the drag coefficient, $C_d$, as well as a specification of the size of the wind farm region (specified in terms of $W$

and $L$), the number of turbines, $N$, and their characteristics (power coefficient $\eta$, cut-in and cut-out velocities, and rotor-swept area $A_{rotor}$).

The implementation of KEBA as well as the evaluations shown in the following section is provided as the Supplementary Material as an Excel spreadsheet.

## 3  Model evaluation

We evaluated KEBA with a set of sensitivity simulations with the WRF regional weather model published by Volker et al. (2017). Volker et al. (2017) evaluated the yield for four different sizes of wind farms, ranging from 25 km$^2$ ("Small") to 114 000 km$^2$ ("X-Large"), with three different turbine spacings (narrow, with an installed capacity density of 11.5 MW km$^{-2}$, intermediate, with 6.38 MW km$^{-2}$, and wide, with 2.9 MW km$^{-2}$) for three wind climates: the central US (Region A), the North Sea (Region B), and the Strait of Magellan (Region C). The scenarios of Volker et al. (2017) as well as their estimated

yields are summarized in Table 2. The KEBA model parameters to evaluate these scenarios are provided in Table 3.



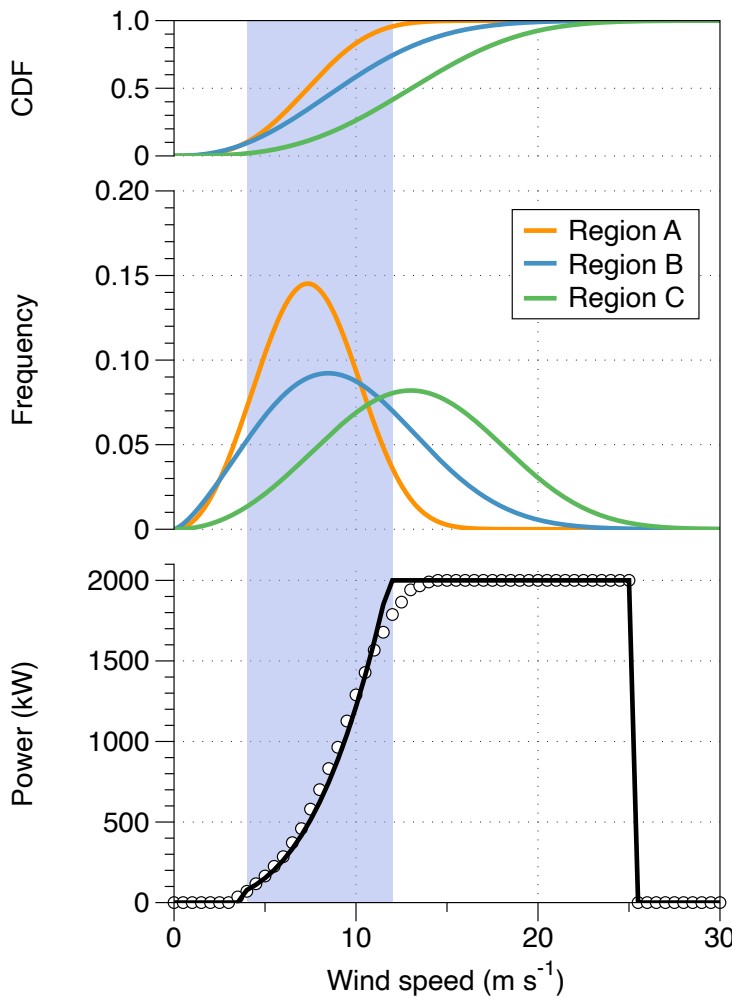

**Figure 2.** Wind forcing and turbine power curve used for the evaluation of KEBA. The top two panels show the frequency distribution of wind speeds for the three regions (A: Iowa, US (orange); B: North Sea (blue); C: Strait of Magellan (green)) considered here including their cumulative probability function (CDF) at the top. The lower panel shows the power curve used here for the Vestas V-80 2 MW wind turbines with a power coefficient of $\eta = 0.44$, with the circles representing the actual power curve of the turbine (obtained from http://www.windpower.net). Both form the inputs of the KEBA evaluation and are based on the data provided in Volker et al. (2017). The blue shaded region reflects wind speeds at which the turbine operates above its cut-in, but below its rated velocity.

## 3.1 Forcing and scenario setup

The wind speed histograms of the three regions considered are shown in Fig. 2a, with median wind speeds of 7.4 m/s, 9.1 m/s, and 13.1 m/s for the three regions. The histograms are represented by Weibull distributions of the form

$$f(v) = \frac{k}{\lambda} \cdot \left(\frac{v}{k}\right)^{k-1} \cdot e^{\left(\frac{v}{\lambda}\right)^k} \tag{15}$$





with values $k = 3.1$ and $\lambda = 8.33$ for Region A, $k = 2.4$ and $\lambda = 10.6$ for Region B, and $k = 3.1$ and $\lambda = 14.7$ for Region C. These distributions were created to resemble closely those of Volker et al. (2017) and have the same medians.

Each scenario considered a set of 2 MW Vestas V-80 turbines ($P_{el,max}$ = 2 MW), with a rotor diameter of $D = 80$ m (yielding a rotor-swept area of $A_{rotor} = 5027$ m$^2$), a cut-in velocity of $v_{min} = 4$ m s$^{-1}$, a cut-out velocity of $v_{max} = 25$ m s$^{-1}$, and a power coefficient of $\eta = 0.44$ when the turbines operate below their capacity. These turbine properties are typically provided by the manufacturer or can be obtained from other data providers such as www.thewindpower.net or www.wind-

5    turbine-models.com. The power curve of the turbine we use here is shown in Fig. 2a and was obtained by fitting the power coefficient $\eta$ to the values provided by www.thewindpower.net.

The scenarios consider four different sizes of wind farms arranged in a square, ranging from about 5 km to 337 km, with three turbine spacings of $5.25D$, $7D$ and $10.5D$, yielding a range of installed capacities in the scenarios from 72 MW to 1293 GW. We evaluated each scenario with KEBA and compare the resulting yields to the reported yields of the WRF simulations.

5    The wind speed histograms shown in Fig.2a are sufficient as inputs for $v_{in}$, that is, these histograms already encapsulate the climatological information of the wind speeds that are needed to evaluate KEBA. As the wind farms are arranged in a square configuration, we did not consider effects of wind direction. For the height $H$, we use typical values for the convective boundary layer heights over land of about 2000 m for Region A and 700 m for the marine climates (Regions B and C). For the drag coefficient we used a value of $C_d \approx 0.001$, which is representative of a relatively smooth surface with a low roughness

10    (such as grassland or an ocean surface) and a reference height of about 100m.

We compare the KEBA estimates also to an estimate in which no wind speed reductions are considered ("isolated" case), so that each turbine operates as if it were an isolated wind turbine. This case is represented in KEBA by a reduction factor of $f_{red} = 1$.

### 3.2    Comparison to Volker et al. (2017) simulations

5    The comparison of KEBA estimates to the estimates by Volker et al. (2017) is shown in Fig. 3 as well as in Table 4. Figure 3 compares yield estimates in absolute terms (Fig. 3a) and in terms of the relative reduction in yield (Fig. 3b) compared to the "isolated" case of what would be expected from turbines that do not experience loss effects due to reduced wind speeds. The comparison shows that KEBA estimates the annual yields very well. The more detailed comparison in terms of the relative yield reduction in Fig. 3b shows that KEBA seems to be better suited at estimating the effect for larger farms, where it shows

10    a closer agreement to the estimates of Volker et al. (2017), while for small wind farms, the yields show a bias towards lower reductions. Using the $n = 36$ simulations as the sample size, a linear regression between KEBA estimates and Volker et al. (2017) yield a correlation of $r^2 = 0.822$ and a slope near one, reflecting the close agreement between the two methods.

The key variable that describes the effect of the kinetic energy removal by the wind turbines is the reduction factor, $f_{red}$, which is given by Eq. 9 for conditions in which the wind turbines operate above the cut-in velocity, but below their rated velocity. This reduction factor is shown in Fig. 4 for the different scenarios, together with the implied reduction in wind speeds (cf. Eq. 8) and yields (cf. Eq. 10). While this reduction factor does not describe all conditions (only those shaded in blue in Fig.



**Table 2.** Scenarios used to evaluate KEBA, as defined in Volker et al. (2017) for three regions, A, B, and C and three turbine spacings: Wide ($10.5D$), Intermediate ($7D$), and Narrow ($5.25D$). The yields are taken from Volker et al. (2017), Table 3.

| Size | $N$ | $W, L$ (in km) | Installed capacity (in GW) | Yield A (in TWh/a) | Yield B (in TWh/a) | Yield C (in TWh/a) |
|------|-----|------|------|------|------|------|
| **Small** | | | | | | |
| Wide | 36 | 5.0 | 0.1 | 0.2 | 0.33 | 0.47 |
| Intermediate | 81 | 5.0 | 0.2 | 0.41 | 0.7 | 1.0 |
| Narrow | 144 | 5.0 | 0.3 | 0.64 | 1.1 | 1.7 |
| **Medium** | | | | | | |
| Wide | 484 | 18.5 | 1.0 | 2.5 | 4.0 | 6.0 |
| Intermediate | 1089 | 18.5 | 2.2 | 4.4 | 7.5 | 12 |
| Narrow | 1936 | 18.5 | 3.9 | 5.9 | 11 | 18 |
| **Large** | | | | | | |
| Wide | 40804 | 169.7 | 82 | 180 | 280 | 440 |
| Intermediate | 91809 | 169.7 | 184 | 280 | 430 | 780 |
| Narrow | 163216 | 169.7 | 326 | 350 | 520 | 1000 |
| **X-Large** | | | | | | |
| Wide | 161604 | 337.7 | 323 | 690 | 1000 | 1700 |
| Intermediate | 363609 | 337.7 | 727 | 1100 | 1600 | 2900 |
| Narrow | 646416 | 337.7 | 1293 | 1300 | 1800 | 3600 |

2), as it does not consider the velocities below the cut-in wind speed or the conditions in which the turbines operate at their rated

5    capacity, it captures the reductions of the different scenarios very well, so that this factor can be used for the interpretation.

Fig. 4 shows how the reduction in effective wind speed becomes greater the larger and denser the wind farm is. Mathematically, this can be seen in the expression for $f_{red}$ (Eq. 9), which decreases with increasing downwind length $L$ of the wind farm. This reduced wind speed is then associated with a reduction factor $f_{red}$ that deviates more and more from the "isolated" case that is represented by $f_{red} = 1$. Note that the actual yield is not always affected by the reduction in wind speed as there

10   are some situations in which wind speeds are above the rated wind speed at which the turbines would operate at their capacity despite the reduction in wind speeds. This can be seen in the estimated yields for Region C, which has a substantial fraction of wind speeds above the rated velocity (above the blue-shaded region in Fig. 2). Hence, the simulated yields for region C are typically less than what is described by this simplified interpretation of Eq. 9.





**Table 3.** KEBA parameters used to evaluate the scenarios of Volker et al. (2017).

| Variable | Specification |
|---|---|
| $v_{in}$ | wind forcing, prescribed by Weibull distribution, see Fig. 2 and Eq. 15. |
| | The following parameters were used for the Weibull distribution: |
| | Region A: $k = 3.1$, $\lambda = 8.33$ |
| | Region B: $k = 2.4$, $\lambda = 10.6$ |
| | Region C: $k = 3.1$, $\lambda = 14.7$ |
| $H$ | Region A (Iowa, land): 2000m |
| | Region B (North sea, ocean): 700m |
| | Region C (Strait of Magellan, ocean): 700m |
| $C_d$ | 0.001 |
| $P_{el,max}$ | 2 MW |
| $A_{rotor}$ | 5027 m$^2$ |
| $\eta$ | 0.44 |
| $v_{min}$ | 4 m s$^{-1}$ |
| $v_{max}$ | 25 m s$^{-1}$ |
| $N$ | dependent on scenario, see Table 2, column $N$. |
| $W$ | dependent on scenario, see Table 2, column $W$. |
| $L$ | dependent on scenario, see Table 2, column $L$. |

### 3.3 Kinetic energy balance diagnostics

Since KEBA is explicitly based on the budgeting of kinetic energy, we can further analyse these scenarios in terms of changes in the energy fluxes within this budget. These terms are approximated here using the mean kinetic energy fluxes ($(\rho/2)v_{in}^3$, Region A: 313.6 W m$^{-2}$; Region B: 742.5 W m$^{-2}$; Region C: 1738.2 W m$^{-2}$) into the regions. The resulting budgets are displayed in Fig. 5 and Table 5. The fluxes are grouped into two terms, the terms that gain kinetic energy by the influx of kinetic energy from upwind areas and from above, and the loss terms, the outflux of kinetic energy, the conversion of kinetic energy into electricity by the wind turbines, as well as the dissipation of kinetic energy within wakes and at the surface. The fluxes are shown in Fig. 5 as relative contributions to the total, so that they are normalized and comparable. The relative contributions depend only on the dimensions of the wind farm volume ($H$, $L$), as well as the drag coefficient ($C_d$) and turbine characteristics ($N$, $\eta$, $A_{rotor}$), but not on the wind climatology of the region. Hence, regions B and C show the same relative contributions and are combined.

The analysis of the kinetic energy budget illustrates how an increasing share of the kinetic energy influx is taken by the turbines and converted into electricity, resulting in less kinetic energy outflux and reduced wind speeds. For the scenarios of small wind farms, essentially all of the kinetic energy supply is provided by the horizontal influx, and the wind farm removes



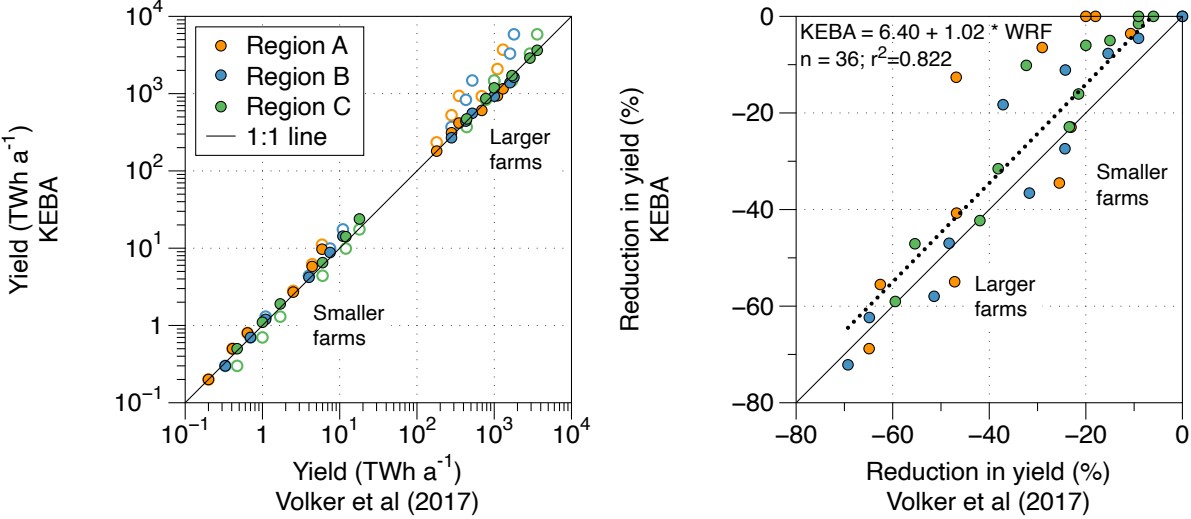

**Figure 3. a.** Comparison of yields for the scenarios of Table 2 estimated by KEBA and as reported by Volker et al. (2017). The open circles represent wind energy estimates that do not take wind speed reductions into account ("isolated", with $f_{red} = 1$). **b.** Relative yield reductions compared to the case of isolated turbines without the wind speed reduction effect. The dotted line reflects a linear regression between the two estimates, with the regression equation provided in the figure.

an insignificant fraction of it. The larger the wind farm region, the more the vertical influx of kinetic energy contributes to the supply of kinetic energy, and wind farms remove an increasingly significant fraction of this supply (yellow bars in Fig. 5). This results in less kinetic energy in the outflux (purple bars in Fig. 5), associated with the unavoidable reduction of wind speeds, which, in turn, is reflected in lower average yields by the turbines.

It is hence this constrained nature of the fluxes that feed the kinetic energy budget of the regional lower atmosphere that encloses the wind farm that results in the diagnosed magnitude of yield reductions. As the estimates by KEBA match those derived from much more complex model simulations by Volker et al. (2017) very well, it is very likely that the same interpretation 5 holds for these simulations, and are thus a realistic representation of the actual dynamics of large wind farms.

### 3.4 Sensitivity to boundary layer height $H$ and drag coefficient $C_d$

We next evaluated the sensitivity of the KEBA estimates to the height of the boundary layer $H$ and the drag coefficient $C_d$. Both of these meteorological parameters are quite uncertain, yet determine how much kinetic energy enters the air volume budgeted by KEBA either in the horizontal or vertical direction. To quantify this sensitivity, we evaluated by how much the 10 yield estimates changed when these two model parameters are varied by ±50% and determined the linear regressions, as shown in Fig. 3b. The estimates changed systematically and yielded weaker (stronger) reductions with greater (smaller) values for $H$ and $C_d$. The regression slopes reflected this change, with the slope reducing to 0.91 and 0.96 when the values of $H$ and $C_d$ were increased, indicating that KEBA would underestimate the yield reduction compared to Volker et al. (2017). When the





**Table 4.** Mean yields estimated for the case of isolated wind turbines (without wind speed reductions), and yield estimates with wind speed reductions by KEBA and by Volker et al. (2017) for three regions, A, B, and C. CF stands for capacity factor ($P_{el,tot}/NP_{el,max}$). The ranges for the "KEBA" and "Volker et al. (2017)" cases refer to the different sizes of the wind farms (Small, Medium, Large, X-Large), with lowest CFs and yields representing the estimates of the largest wind farms.

| Scenario | | "Isolated" | | "KEBA" | | "Volker et al. (2017)" | |
|---|---|---|---|---|---|---|---|
| | $nP_{el,max}$ | CF | Yield | CF | Yield | CF | Yield |
| | MW km$^{-2}$ | (%) | (W m$^{-2}$) | (%) | (W m$^{-2}$) | (%) | (W m$^{-2}$) |
| **Region A** | | | | | | | |
| Wide | 2.83 | 32.7 | 0.9 | 21.2 - 32.4 | 0.6 - 0.9 | 24.4 - 31.7 | 0.6 - 0.9 |
| Intermediate | 6.38 | 32.7 | 2.1 | 14.5 - 32.0 | 0.9 - 2.0 | 17.3 - 28.9 | 1.1 - 1.9 |
| Narrow | 11.34 | 32.7 | 3.7 | 10.0 - 31.5 | 1.1 - 3.6 | 11.5 - 25.4 | 1.3 - 2.9 |
| **Region B** | | | | | | | |
| Wide | 2.83 | 51.7 | 1.5 | 32.1 - 50.9 | 0.9 - 1.4 | 35.3 - 52.3 | 1.0 - 1.5 |
| Intermediate | 6.38 | 51.7 | 3.3 | 21.0 - 50.0 | 1.3 - 3.2 | 25.1 - 49.3 | 1.6 - 3.2 |
| Narrow | 11.34 | 51.7 | 5.9 | 13.8 - 48.6 | 1.6 - 5.5 | 15.9 - 43.6 | 1.8 - 5.0 |
| **Region C** | | | | | | | |
| Wide | 2.83 | 78.4 | 2.2 | 59.7 - 77.8 | 1.7 - 2.2 | 60.0 - 74.5 | 1.7 - 2.2 |
| Intermediate | 6.38 | 78.4 | 5.0 | 44.1 - 77.1 | 2.8 - 4.9 | 45.5 - 70.5 | 2.9 - 4.6 |
| Narrow | 11.34 | 78.4 | 8.9 | 31.0 - 76.0 | 3.5 - 8.6 | 31.8 - 67.4 | 3.6 - 7.8 |

values for $H$ and $C_d$ decreased, the slope increased to 1.17 and 1.10, reflecting an overestimation of the yield reduction. Yet, the slope changed substantially less than the imposed change of 50% to $H$ and $C_d$, indicating that the KEBA estimates are relatively insensitive to these two parameters.

### 3.5 Sensitivity to downwind length $L$

5    The sensitivity to downwind length $L$ is more of scientific interest, as it does not reflect a model uncertainty because it is specified by the evaluated scenario. This sensitivity is of interest because it links the high yields and efficiencies of individual turbines and small wind farms to the low, large-scale wind energy potential of less than 1 W m$^{-2}$.

   To evaluate this sensitivity, we use the meteorological forcing of Region B (North Sea) and evaluate how the KEBA reduction factor, $f_{red}$, the capacity factor (i.e., the average yield of a turbine divided by its capacity, $P_{el}/NP_{el,max}$), as well as the kinetic energy influx per surface area of the wind farm change with $L$. This sensitivity is shown in Fig. 6 for the three installed capacity

5    densities considered above. Note that a downwind length of $L = 0$ represents the case of an isolated wind turbine. In this case, the reduction factor in KEBA is $f_{red} = 1$. In the specified meteorological forcing, a wind turbine would achieve a capacity factor of 54.1%, which would be provided solely from the horizontal influx of kinetic energy.





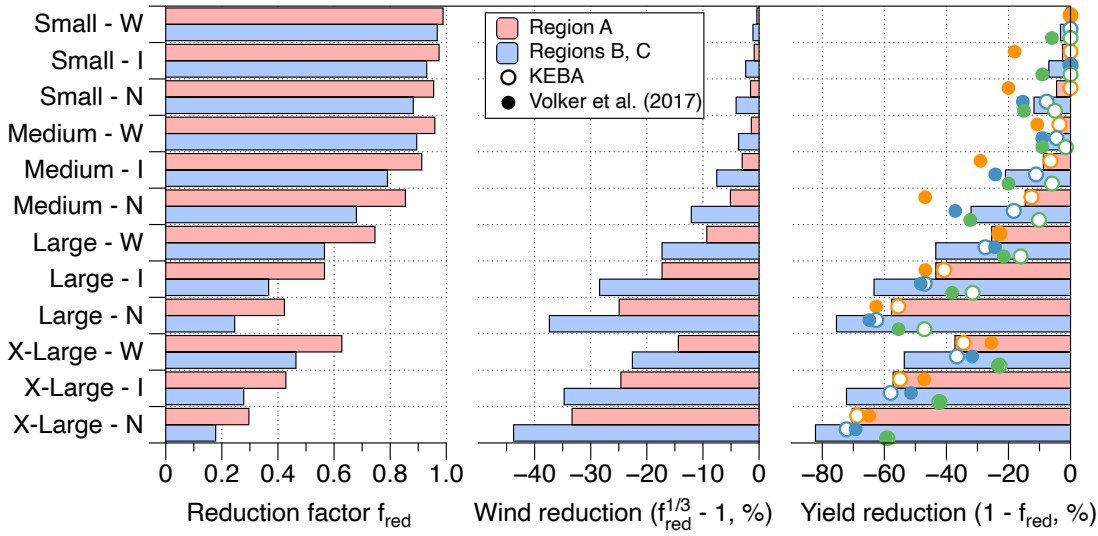

**Figure 4.** The value of the reduction factor $f_{red}$ for the case in which $v < v_{rated}$ (Eq. 9) for the different scenarios as well as the implied relative reduction in wind speed, $f_{red}^{1/3} - 1$, and the relative reduction of yield, $1 - f_{red}$, derived directly from the value of $f_{red}$. The circles in the right panel refer to the actual estimates that include the information of the wind speed histograms (open circles: KEBA; full circles: Volker et al. (2017)), with the color coding as in Fig. 3.

As $L$ increases and the horizontal influx of kinetic energy gets depleted by the wind turbines, the reduction factor drops, and so does the capacity factor. The drop of the reduction factor is relatively fast, reducing to a value of 0.5 within 260 km, 82

10  km, or 42 km for an installed capacity density of 2.8, 6.4, or 11.3 MW km$^{-2}$ for the cases of wide, intermediate, and narrow turbine spacings. This spatial scale is linked to the length scale, $L_d$, associated with an exponential decay of the horizontal kinetic energy input and is described by $L_d = H/(2C_d + n\eta A_{rotor})$.

For very long downwind lengths ($L \rightarrow \infty$), the yield per surface area in KEBA reaches a limiting value of $(n\eta A_{rotor})/(2C_d + 3/2n\eta A_{rotor}) \cdot C_d \rho v_{in}^3$, which is less than 2/3 of the natural frictional dissipation rate of the region. This limit is then again

15  consistent with the global-scale energetics of the large-scale atmospheric circulation, which generates about 2 W m$^{-2}$ of kinetic energy and dissipates about 1 W m$^{-2}$ within the boundary layer. The North Sea region is windier than the global mean with a frictional dissipation of about 2 W m$^{-2}$, so that KEBA would yield a maximum generation of 1.33 W m$^{-2}$. The drop in yields of wind farms from small to very large scales thus reflects the transition from the dominant contribution by a high local horizontal kinetic energy flux to a low global generation and dissipation rate of kinetic energy.

## 3.6 Limitations

The KEBA approach is, clearly, extremely simple and neglects many complicating factors, such as the role of stability, different drag coefficients, variations in boundary layer height, or the wind direction. KEBA can nevertheless reproduce the wind energy yields and their reductions in large wind farms simulated by the much more complex WRF simulations of Volker et al. (2017).



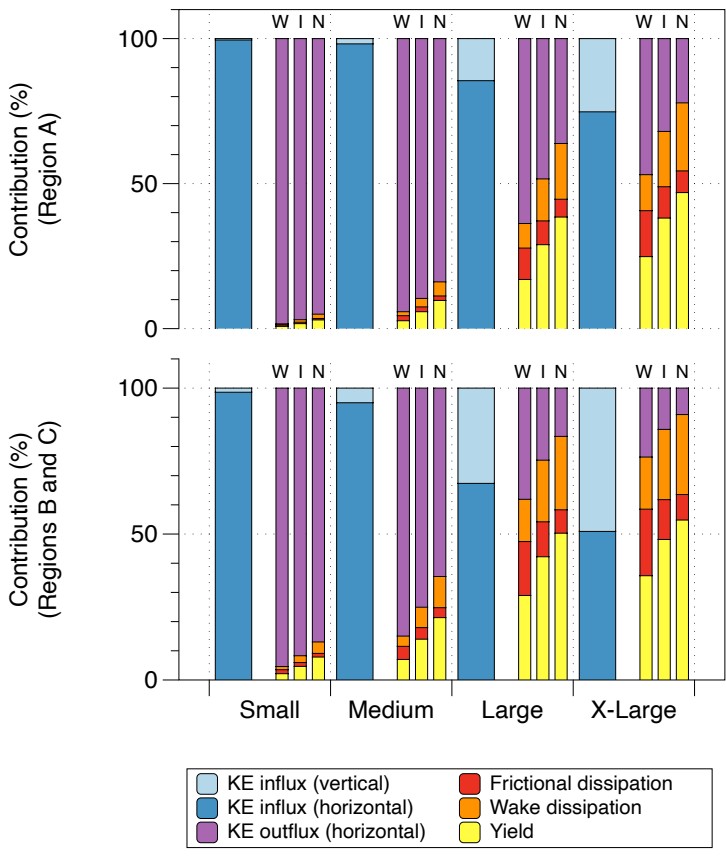

**Figure 5.** Diagnosed terms of the kinetic energy budget, with the relative contributions to the influxes of kinetic energy (KE) shown by the left column and the conversion to electricity, dissipation and outflux of kinetic energy shown by the three right columns (for three different turbine spacings: W: Wide; I: Intermediate; N: Narrow). The horizontal and vertical contributions to the KE influx are shown by dark and light blue respectively. The yield (conversion to electricity) is shown in yellow and the dissipation by surface friction (red) and wake turbulence (orange) and outflux of kinetic energy (purple). The plots show the KE budgets for the scenarios of wind farms of four different areas (S, M, L, and XL) for region A (Central US, land, **a.**), and for regions B and C (North Sea and Straight of Magellan, ocean, **b.**). The colors of the bars match the arrows shown in Fig. 1.

As we deal with climatological estimates, it would seem that the effects of stable and unstable conditions may average out. It therefore suggests that KEBA works well because the most relevant factor is the depletion of the horizontal flow of kinetic

energy. This horizontal flow is prescribed by the wind conditions and is thus insensitive to stability and the value of the drag coefficient. Variations in boundary layer height, captured in our approach by the height $H$, play a more prominent role as these directly affect the total horizontal inflow of kinetic energy. Wind direction in our evaluations probably did not play a large role because we considered simple, artificial square layouts of wind farms. In future applications, it would be insightful and





**Table 5.** Estimated kinetic energy influxes for the different scenarios and wind farm sizes in comparison to the estimated yields. The ranges for the "KEBA" and "Volker et al. (2017)" cases refer to the different turbine spacings (Narrow, Intermediate, Wide) with lowest CFs and yields corresponding to the narrowest spacing and highest turbine densities.

|  | Small | Medium | Large | X-Large |
|---|---|---|---|---|
| **Region A** | | | | |
| Horizontal influx (GW) | 3.14 | 11.60 | 106.44 | 211.81 |
| Vertical influx (GW) | 0.02 | 0.21 | 18.06 | 71.53 |
| Yield (GW) | 0.02 - 0.10 | 0.32 - 1.15 | 21.10 - 47.88 | 70.22 - 132.87 |
| **Region B** | | | | |
| Horizontal influx (GW) | 2.60 | 9.62 | 88.20 | 175.52 |
| Vertical influx (GW) | 0.04 | 0.51 | 42.77 | 169.35 |
| Yield (GW) | 0.06 - 0.21 | 0.71 - 2.16 | 37.91 - 65.85 | 123.18 - 188.96 |
| **Region C** | | | | |
| Horizontal influx (GW) | 6.08 | 22.51 | 206.48 | 410.89 |
| Vertical influx (GW) | 0.09 | 1.19 | 100.11 | 396.45 |
| Yield (GW) | 0.07 - 0.25 | 0.87 - 2.64 | 46.24 - 80.32 | 150.26 - 230.51 |

potentially necessary to evaluate the effects of these aspects on simulated wind speed reductions and yields. The estimate by

KEBA could help to set a baseline for such evaluations.

Our KEBA approach also only crudely describes the wakes that develop directly behind wind turbines through the wake dissipation term, $D_{wake}$ (Eq. 7) and does not resolve differences in yields within the same wind park. For these wake effects, approaches are already available to capture these (e.g., Frandsen et al., 2006; Barthelmie et al., 2010; Emeis, 2010). Future extensions could aim to combine such approaches to yield a model that can not only estimate regional wind energy potentials,

but also variations within wind farms. On the other hand, it would seem that more complex numerical simulations of wind farm effects would benefit from an analysis of the kinetic energy budget.

In its present form KEBA can adequately capture wind energy resource estimates at the regional scale and, as such, can inform the planning and policy development regarding the future expansions of wind energy. By being implemented in a spreadsheet, it can quickly estimate the yields of different scenarios in a transparent and reproducible way, given the prescribed

wind conditions of the region. This was, for instance, recently demonstrated in a study that evaluated German wind energy scenarios for offshore wind energy in the North Sea, where the KEBA model was used (Agora Energiewende et al., 2020).

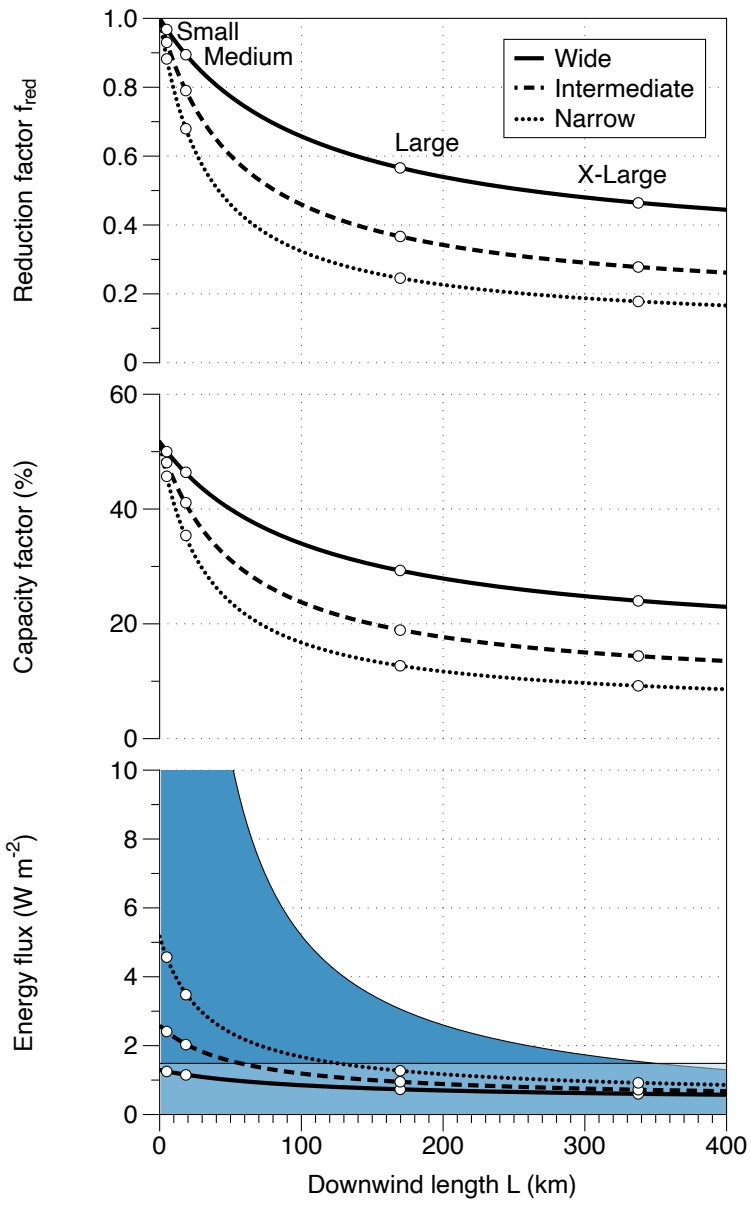

**Figure 6.** KEBA sensitivity to downwind length $L$ for Region B (North Sea). The panels show the KEBA reduction factor $f_{red}$ as a function of downwind length for three turbine spacings, the associated reduction in capacity factor ($P_{el}/NP_{el,max}$, and the supply of kinetic energy by the horizontal influx from upwind areas (dark blue) and by vertical mixing (light blue) as well as the yields by the wind farms (lines). The circles represent the different sizes of wind farms considered in the comparison with Volker et al. (2017).





# 4 Conclusions

We presented a model to estimate wind energy resource potentials at the regional scale that explicitly accounts for wind speed reductions caused by the wind turbines. This formulation yielded analytical solutions to estimate these wind speed reductions and the associated mean yields of the wind farms. We compared this formulation to a set of sensitivity simulations with the WRF regional weather forecasting model by Volker et al. (2017) and found that KEBA can adequately reproduce yield reductions.

The modelling of the kinetic energy budget thus provides valuable insights for estimating wind speed reductions and wind energy resource potentials at the regional scale, but also at a more general level in terms of understanding the impacts that large-scale wind energy use has on the atmosphere. While our approach can be extended in future work to address some of the shortcomings, it seems that an explicit analysis of kinetic energy fluxes would be informative and provides valuable information. In its present form, KEBA seems well suited to provide first-order estimates of wind energy resource potentials at the regional scale that are based on atmospheric physics.

*Code and data availability.* The KEBA implementation is provided in an Excel spreadsheet available as Supplemental Material. All data used to evaluate KEBA is contained in the Excel spreadsheet.

*Author contributions.* AK and LM designed the study, performed the analysis and wrote the paper; AK performed the KEBA simulations. The authors declare no competing financial interests.

*Competing interests.* The authors declare that they have no competing interests.





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
