# Peer review of "The Kinetic Energy Budget of the Atmosphere (KEBA) model 1.0: A simple yet physical approach for estimating regional wind energy resource potentials that includes the kinetic energy removal effect by wind turbines"

_Geoscientific Model Development, 2020_

## Referee Comment (RC1) · Anonymous Referee #1 · 25 May 2020

This paper is very well written and designed. It addresses an important topic for wind industry: How to simply, yet physically model the energy budget in the atmospheric boundary layer. Perfectly achieved with KEBA ! The model is validated against much more sophisticated WRF simulations and - very important - the authors also address its shortcomings and limitations. I recommend publication after the correction of some technical mistakes. - Fig.1 and text: The figure uses different terms than in the text. The text versions should be applied, e.g., "surface friction" instead of "turbulent dissipation"

and "wake turbulence" instead of "wake dissipation". - Fig. 1: the energy fluxes in the figure are no fluxes but flux densities; the correct fluxes are described in the equations on p. 6 - eq. 9 and 11 for f_red : since one wind turbine yields in f_red = 1, n= (N-1)/WL in (9) and (N-1) in (11) should be used.

———————————————

---

## Referee Comment (RC2) · Daniel Kirk-Davidoff (Referee) · 17 Jul 2020

In this manuscript, Kleidon and Miller extend their work on the ultimate recoverable wind resource to a practical tool to estimate that resource using data available from reanalysis data sets, and show that it performs reasonably well compared to a reasonably detailed modeling effort using WRF. The paper is clearly written, and the authors have provided convenient access to the details of the model. The basic result that the

realistically achievable wind resource (as determined by WRF!) is clearly related to the boundary layer winds in the absence of the wind farm is encouraging, though perhaps not surprising given the many years of experience the wind industry has in performing and validating pre-construction resource estimates.

I am concerned, however, that the particular range of wind turbine densities studied may make the paper subject to misinterpretation, so I'd like the authors to include a paragraph or so relating the chosen densities (which I recognized they've inherited from Volker et al (2017)) to real-world densities. Typical large wind farms have densities of less than 4.5 MW km$^{-2}$, averaging closer to 3 MW km$^{-2}$ (see, for example, Denholm et al., 2009, "Land-Use Requirements of Modern Wind Power Plants in the United States"). This is quite close to the least dense "wide" category considered here (2.8 MW km${-2}$), and far from the "intermediate" (6.4 MW km${-2}$) or "narrow" (11.3 MW km${-2}$) categories. A novice, reading those category labels might think that "intermediate" density corresponded to a typical real-world density. Further, since the estimates from the KEBA model seem intended to give a quick sense of the potential generation from wide areas, it's probably worth noting that a "wide" density of wind farms, installed over Iowa, would imply an installed capacity of 400 GW, compared to an actual installed capacity of 10.6 GW as of 2019.

Additionally, I think it would be helpful to re-express the values shown in Table 2 in TWh/a in units of W/m$^2$, so that the deviation from the simplest hypothesis of a fixed limit in terms W/m$^2$ is made obvious. Finally, I think a clearer description of how exactly the various input parameters ($v_{in}$, and $H$, for example) are derived from the WRF model (e.g. from which height in the model is $v_{in}$ taken) would be very helpful to the reader.
* * *

---

## Author Response (AR1)

**The KEBA model 1.0: Revisions of manuscript**

Axel Kleidon and Lee Miller 28 July 2020

Dear Editor,

we submit the revised version of our manuscript for possible publication in Geoscientific Model Development.

In the revision, we implemented the changes as described in our response to the reviews (separate document in the discussion forum, https://doi.org/10.5194/gmd-2020-77-AC1). We submitted one version of the manuscript where the changes are marked in blue. These changes are:

Page 2, line 11-12: Added a sentence in response to reviewer comment #2-1. Page 3, Figure 1: Revised labeling of fluxes and adjusted expressions in response to reviewer comments #1-1 and #1-2.

Page 5, Table 1: Adjusted description of terms  $D_{fric}$  and  $D_{wake}$  in response to reviewer comment #1-2.

Page 7, line 7 and Eq. 11: Changed equations in response to reviewer comments #1-3.

Page 10, lines 2-3, 15-18: Added information about  $v_{in}$  and H in response to reviewer comment #2-3.

Page 11, Table 2: Added information of yields in W m-2 in response to reviewer comment #2-2. Page 12, line 7: Corrected terminology.

Page 18, line 17- Page 19, line 14: Extended a paragraph to better set the capacity densities of the manuscript in context of the real world and energy scenarios in response to reviewer comment #2-1.

We think that with these changes, we addressed all of the concerns of the reviewers.

Thank you and kind regards, Axel Kleidon

We thank both reviewers for their very positive and constructive reviews. In the following, we provide a point-by-point response to both reviews combined, because not many comments have been made. We put the reviewer comments in bold type, followed by our response.

Reviewer 1: This paper is very well written and designed. It addresses an important topic for wind industry: How to simply, yet physically model the energy budget in the atmospheric boundary layer. Perfectly achieved with KEBA ! The model is validated against much more sophisticated WRF simulations and - very important - the authors also address its shortcomings and limitations. I recommend publication after the correction of some technical mistakes.

Response: Thank you very much for the kind, positive words!

Comment 1-1: Fig.1 and text: The figure uses different terms than in the text. The text versions should be applied, e.g., "surface friction" instead of "turbulent dissipation" and "wake turbulence" instead of "wake dissipation".

Response: Yes, we agree and will adjust Fig. 1 (and Table 1) accordingly.

Comment 1-2: Fig. 1: the energy fluxes in the figure are no fluxes but flux densities; the correct fluxes are described in the equations on p. 6

Response: Yes, we agree and will adjust Fig. 1 accordingly.

Comment 1-3: eq. 9 and 11 for  $f_{red}$ : since one wind turbine yields in  $f_{red} = 1$ , n = (N-1)/WL in (9) and (N-1) in (11) should be used.

Response: Excellent point, well spotted! Yes, we will correct this.

Reviewer 2: In this manuscript, Kleidon and Miller extend their work on the ultimate recoverable wind resource to a practical tool to estimate that resource using data available from reanalysis data sets, and show that it performs reasonably well compared to a reasonably detailed modeling effort using WRF. The paper is clearly written, and the authors have provided convenient access to the details of the model. The basic result that the realistically achievable wind resource (as determined by WRF!) is clearly related to the boundary layer winds in the absence of the wind farm is encouraging, though perhaps not surprising given the many years of experience the wind industry has in performing and validating pre-construction resource estimates. **Response:** Thank you for the succinct description. However, we do not quite agree with the statement that this is "perhaps not surprising". In our own experience from discussions with the wind energy industry and energy policy community in Germany and Europe, the reduced wind energy potentials at larger scales due to the wind speed reductions that are captured by KEBA is not at all established, so we think that the KEBA model certainly provides a novel tool (see also response to the following, first comment).

Comment 2-1: I am concerned, however, that the particular range of wind turbine densities studied may make the paper subject to misinterpretation, so I'd like the authors to include a paragraph or so relating the chosen densities (which I recognized they've inherited from Volker et al (2017)) to real-world densities. Typical large wind farms have densities of less than 4.5 MW km-2, averaging closer to 3 MW km-2 (see, for example, Denholm et al., 2009, "Land-Use Requirements of Modern Wind Power Plants in the United States"). This is quite close to the least dense "wide" category considered here (2.8 MW km-2), and far from the "intermediate" (6.4 MW km-2) or "narrow" (11.3 MW km-2) categories. A novice, reading those category labels might think that "intermediate" density corresponded to a typical real-world density. Further, since the estimates from the KEBA model seem intended to give a quick sense of the potential generation from wide areas, it's probably worth noting that a "wide" density of wind farms, installed over lowa, would imply an installed capacity of 400 GW, compared to an actual installed capacity of 10.6 GW as of 2019.

**Response:** We agree that we can motivate these densities better and the context in which KEBA becomes important. While current capacity densities are at the lower end of what is being considered, there are, however, also plenty of studies in the scientific literature and policy scenarios that consider much higher capacity densities over large areas. For example, the recent study by Enevoldsen et al. (2019, Energy Policy 132: 1092-1100) assume a capacity density of 10 MW km-2 over half the area of Europe,

СЗ

which is close to the "Narrow" density considered in Volker et al. (2017). Also, energy scenarios for offshore wind energy in Germany by the year 2050 assume up to 70 GW installed over 2800 km-2, (an area that falls between the "Medium" and "Large" size considered in the manuscript), resulting in a mean capacity density of 25 MW km-2. In this sense, we do not think that the capacity densities are unrealistic, at least not in the context of the scientific literature and some renewable energy scenarios.

In the revision, we will make these points more explicit by including a sentence with a the reference to current capacity densities and the Enevoldsen et al. (2019) study in the introduction as two examples of capacity densities for clarification. Also, we will add a few sentences to extend the description of the Agora study in the discussion section to illustrate the reasonability of the assumed values as well as the implications.

Comment 2-2: Additionally, I think it would be helpful to re-express the values shown in Table 2 in TWh/a in units of  $W/m^2$ , so that the deviation from the simplest hypothesis of a fixed limit in terms  $W/m^2$  is made obvious.

Response: Yes, we agree and will modify Table 2 accordingly.

Comment 2-3: Finally, I think a clearer description of how exactly the various input parameters ( $v_{in}$ , and H, for example) are derived from the WRF model (e.g. from which height in the model is  $v_{in}$  taken) would be very helpful to the reader.

**Response:** The wind input is taken from the histograms provided in Fig. 1 of Volker et al. (2017). For the boundary layer heights, we used relatively rough, typical values drawn from the literature (Seidel et al., JGR, 2009 for land from radiosoundings, Peña et al., JGR, 2013 for the North sea). We will add these references in the revision.

[revised manuscript text omitted]

---

## Author Response (AR2)

**The KEBA model 1.0: Revisions of manuscript**

Axel Kleidon and Lee Miller
24 August 2020

Dear Editor,

we submit the revised version of our manuscript which includes the minor comments from you:

- We added labeling of the figures (a, b, c, ...) to the panels of Figures 2, 3, 5 and 6
- We adjusted the precision in Tables 2, 4 and 5, following the precision given in Volker et al. (2017), which we mention in the caption of Table 2.

The modifications are marked in blue in the uploaded PDF.

We think that with these changes, we addressed all of your comments.

Thank you and kind regards,
Axel Kleidon